# Experimental Analysis of the Influence of Urban Morphological Indices on the Urban Thermal Environment of Zhengzhou, China

**Xuefan Zhou and Hong Chen ***

School of Architecture and Urban Planning of Huazhong University of Science and Technology, Wuhan 430074, China; xuefanzhou@hust.edu.cn
* Correspondence: chhwh@hust.edu.cn

**Abstract:** Summer extreme high-temperatures occur frequently in large cities; urban spatial form is the primary factor affecting the urban thermal environment. Thus, planning and arranging urban spaces is a key approach to regulating urban microclimates. Studies into how urban spatial forms influence the formation of urban microclimates have been carried out for multiple cities in warm and hot regions; however, few studies of this kind have been carried out for cities in cold regions. In this study, we analyze Zhengzhou, a city located in a cold region of China, using summer 2017 measurement data to determine why high temperatures develop in cold areas. We investigated how temperature and humidity vary during the morning, at noon, and in the evening given different land use properties (commercial and residential) and different spatial forms (building height, building density, green coverage rate, and plot ratio); we then studied the correlation between urban spatial form and the urban thermal environment. Our research results indicate that the commercial district's thermal microclimate was related to PR and BH in the afternoon and GCR in the morning and at night. In the residential district, the key urban morphology factors related to its thermal microclimates were BD, PR, and GCR during almost the whole day.

**Keywords:** urban form; land use; mobile measurement; urban thermal environment

## 1. Introduction

The current warming trend in global climate is significant, and summer extreme high-temperature weather events occur frequently in cities, posing a great threat to the health of urban residents [1]. Multiple cities located in the warm climate zone (hot regions) have been studied in detail and from multiple perspectives to determine how urban high temperatures are generated [2], the extent to which various factors influence urban high temperatures, and what can be done to improve urban temperatures [3]. However, few relevant studies of cities in cold climates have been conducted [4]. Many meteorological data indicate that the warming trend in cold-climate cities is more significant than those in warm and hot areas [5,6]. Zhou et al. studied 32 cities in China, and indicated that the Urban Heat Island Intensity (UHII) differed substantially between day and night, with distinct climate-driven patterns, characterized by a higher UHII in the southeastern parts of China during the day, and in the northern regions of the country at night compared with other regions. In particular, the northeast, with humid-cold climate, experienced the most intense UHII across China [7]. Moreover, because people who live in cold climates are not adapted to high summer temperatures, the resultant health threat to residents is more serious [8]. Therefore, before we identify ways to alleviate high summer temperatures in the cold-climate areas, we must first analyze how urban high temperatures form in the cold areas and identify the primary factors influencing urban high temperature variations.

Urban populations are rapidly expanding; this explosive growth causes changes in the heat gains and losses of cities, resulting in the formation of special urban microclimates [9].

In particular, urban growth results in the urban heat island effect, a phenomenon where the urban air temperature is generally higher than the air temperature in the surrounding suburban area. The urban heat island effect is the concentrated manifestation of a more general urban high temperature trend. Many factors drive the development of an urban heat island, including high artificial heat emissions generated by high-density population activity [10], increases in impermeable surfaces, and artificial underlying surfaces with high regenerative heat properties [11]. Urban forms are becoming more and more compact [12], buildings are getting taller [13], and the green coverage rate is shrinking [14]. Previous studies show that city's geographic location, elevation height, urban size, and urban land uses, as well as solar radiation, wind, precipitation, Sky View Factor (SVF), building density, greenery, albedo, and water area may greatly affect urban air temperature [15]. However, there are still large differences in the factors causing different urban heat island effects among different cities.

Gerçek et al. [16] used remote sensing data to analyze the cause of the heat island in Izmir, Turkey, and found that the intensity of the nighttime heat island was primarily affected by traffic heat discharge, while the intensity of the daytime heat island was directly related to sky rate (Sky View Factor), green rate, and the reflectance and absorptivity of building materials. Middel et al. used ENVI-met [17] and RayMan to simulate the thermal environment in Phoenix, AZ, USA, a city with a subtropical desert climate, and found that the city's compact urban morphology largely mitigated the heat island problem. In particular, sky view factor is the primary driver of the thermal environment in hot, dry regions. Zhao et al. [18] used remote sensing to analyze the ground surface temperature in Shanghai from 1984 to 2014 and found that, when urban compactness is less than 0.15, there is no obvious influence on the urban heat island; however, urban morphology has a significant influence on the thermal environment when the urban 63 compactness is higher than 0.15. Ali et al. [19] studied the heat island problem in Bhopal, India and found that the temperature difference between a region with a green rate of 0.5 and an adjacent urban area reaches approximately 6 °C. Sharmina et al. [20] used the ENVI-met simulation to study the influence of street height-to-width ratio (H/W) and sky view factor on the intensity of the heat island effect in Dhaka, the capital of Bangladesh. Adachi et al. [21] used the mesoscale weather simulation tool Weather Research and Forecasting Model (WRF) to simulate and analyze the thermal environment in Tokyo, Japan. They found that the compact city center plays a role in ameliorating the heat island effect. This indicates that compact spatial form will play a significant role in reducing the heat island effect in and around cities by the occlusion effect. Alobaydia et al. [22] used ENVI-met to simulate the thermal environment in Baghdad, Iraq, and found that, when the street height-to-width ratio (H/W) is higher, the ground surface temperature and air temperature in cities in dry and hot climates are lower. A similar conclusion was reached in Riyadh, and Bakarmana and Changa [23] used measurements to find that street temperatures of streets with a relatively high height-to-width ratio (H/W = 2.2) are greater than those of streets with a relatively small height-to-width ratio (H/W = 0.42) by approximately 10%. Xu et al. [24] used remote sensing to collect data and established the urban 3D model of Kowloon District, Hong Kong; they studied the influence of building density, building height, plot ratio, upwind area ratio, and sky view factor on the heat island effect in humid and hot regions. Similarly, Liu et al. [25] used observational data to study the degree of influence of sky view factor, street aspect ratio, building area, permeable pavement ratio, nonpermeable pavement ratio, average height of building, geographical roughness grade, surface heat absorption, and surface reflectivity on the heat island effect in Shenzhen City. They found that the geographical roughness grade most significantly impacted the thermal environment in Shenzhen City, followed by street aspect ratio, surface heat absorption, and average building height. The influence of other parameters on the thermal environment in Shenzhen City is limited. Yang et al. [26] used remote sensing to analyze the reason for the heat island effect in Chongqing City and found that different land use types responded differently to ground surface temperature. The ground surface temperature of construction

land and unused land was greatest, followed by that of grassland, arable land, forestland, and water. The study also found that the intensity of the urban heat island is obviously negatively correlated with vegetation coverage. The greater the vegetation coverage, the lower the intensity of the heat island effect. When vegetation coverage is greater than 50%, the heat island phenomenon essentially disappears, and the average temperature difference inside and outside the urban green space can reach approximately 2.2 °C. The larger the green area, the stronger the temperature difference compared to the surrounding area.

In summary, urban land attributes have an absolute effect on the formation of the urban microclimate (the first category of landscape parameters that influence urban microclimates is land cover features) [27]. This is because a difference in a specific land attributes directly causes a change in the urban morphology and artificial heat emissions in the corresponding areas. Therefore, many studies further investigated the influence of urban morphology and artificial heat emissions on the urban thermal environment given land attributes [28]. The factors that significantly influence the urban thermal environment also include green coverage rate, building height, building density, plot ratio, and sky rate, which can be represented by multiple planning and design indexes (landscape design parameters) that characterize the urban morphology [29]. However, the studies of urban morphology impact on urban heat island are mainly explored in hot climate regions. Just a few of the studies are focused on extremely cold regions. For example, Oke and Maxwell discovered that the heat island of Montreal and Vancouver grew most rapidly following sunset because of much stronger rural cooling [5]. Varentsov et al. discovered that the UHI in Moscow shows a strong diurnal variation in terms of intensity and vertical extent between daytime ($\approx$0.5 K/$\approx$1.5 km) and nighttime (>3 K/$\approx$150 m) [6]. From these studies, we may notice that urban morphology impact of these cold region cities is quite different from hot climate region cities. Whether cold-climate urban micro-climates also generate increasingly serious urban heat island effects due to the influence of the above planning and design indexes is the subject of this paper. We study what causes summer high-temperature weather in cities located in traditionally cold climates and which planning and design indexes have the strongest and weakest correlations with the urban thermal environment.

We take the city of Zhengzhou in a typically cold area of China as an example and use observed data from the summer of 2017 to analyze the primary reasons for the formation of high-temperature weather in cold areas. We investigate how temperature and humidity vary in the morning, at noon, and in the evening for different land use properties (commercial and residential) and regions with different spatial forms—building height (BH), building density (BD), green coverage rate (GCR), and plot ratio (PR). Based on a nonlinear regression model, we study the correlation between urban spatial form and the urban thermal environment, and we identify the main factors influencing the summer thermal environment in cold areas.

## 2. Study Area and Methodology

### 2.1. Study Area

Zhengzhou is located in central China and is the capital of Henan Province. It is located in the southern portion of the North China Plain in the lower reach of the Yellow River. It lies between 112°42′ E and 114°14′ E, and 34°16′ N and 34°58′ N (Figure 1). According to the dataset supplied by Zhengzhou government website (zhengzhou.gov.cn, accessed on 21 June 2021), the total area of Zhengzhou City is 7446 km². Until the end of 2015, Zhengzhou city covered six municipal districts and one county and representatively administered five county-level cities. The permanent resident population is 9569 million. This region experiences a temperate monsoon climate. Southeastern winds dominate in the summer, while northwestern winds dominate in the winter; the average wind velocity is approximately 2.4 m/s. The region experiences four distinct seasons, including a relatively cold winter. Therefore, it is classified as a cold area according to the Chinese Architectural Climate Zoning Standard. Although the historical average maximum summer air temperature in Zhengzhou is approximately 30 °C (Figure 2), the acceleration of urbanization in recent years has resulted in a taller, denser urban spatial form. Artificial heat emissions

have increased, resulting in maximum summer air temperatures of approximately 38 °C to 40 °C, a summer extreme maximum temperature phenomenon.

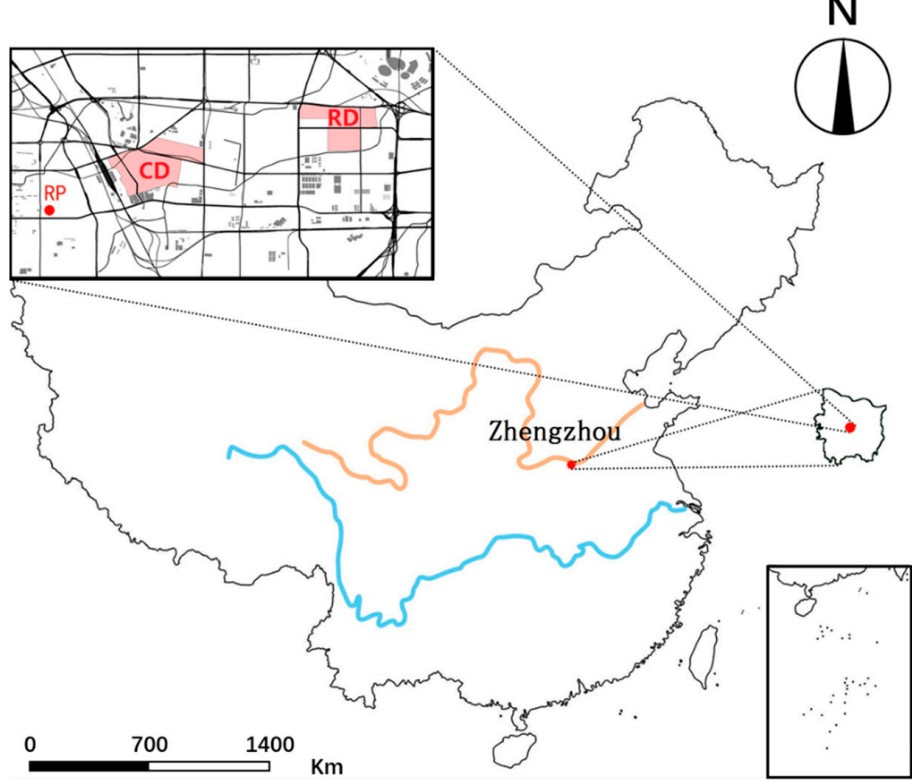

**Figure 1.** Locations of Zhengzhou city, sampling commercial district (CD), residential district (RD), and reference point (RP).

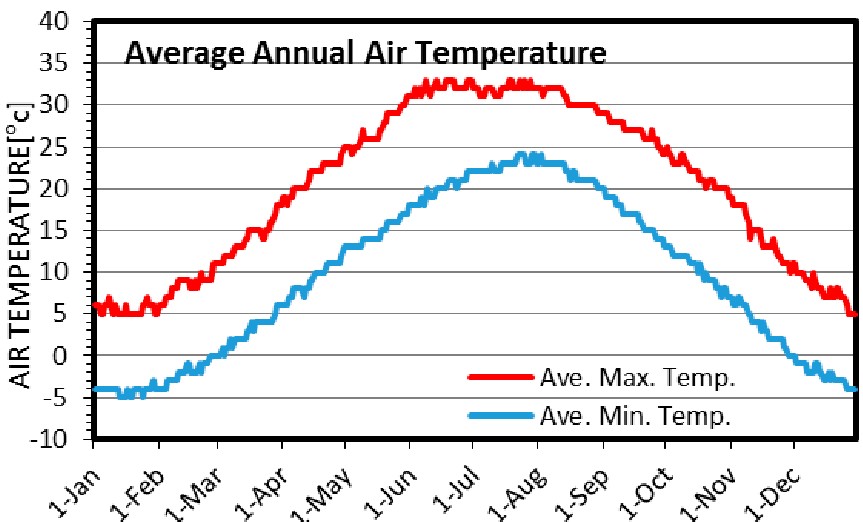

**Figure 2.** Average annual maximum air temperature and minimum air temperature of Zhengzhou (data source from http://www.weather.com.cn/forecast/history.shtml?areaid=101180101&month, accessed on 21 June 2021).

To further illustrate the influence of urban spatial form on the thermal environment based on different land use properties in Zhengzhou, we selected a typical commercial district and a typical residential district for evaluation and established a fixed reference point. Those three locations are shown in Figure 1.

In this study, we selected the Er-qi commercial district to represent a typical commercial district in Zhengzhou. The circumference of the commercial district is approximately 7.4 km. The average building height is approximately 16 m. Roads are generally 4-lane arterials that are approximately 16 m wide. In the commercial district, we sampled 27 circular areas, each with a radius of 100 m; these 27 sampling circles' BD range from 10.04 to 60.18%, with an average of 31.92%, PR is from 0.56 to 3.96, with an average of 2.02, BH ranges from 7.6 m to 28 m, with an average of 16.87 m, and GCR ranges from 0.72 to 41.81%, with an average of 15.67%; the locations of these sample sites are shown in Figure 3A. The mean specific indexes of urban spatial form for each sampling circle are shown in Table 1.

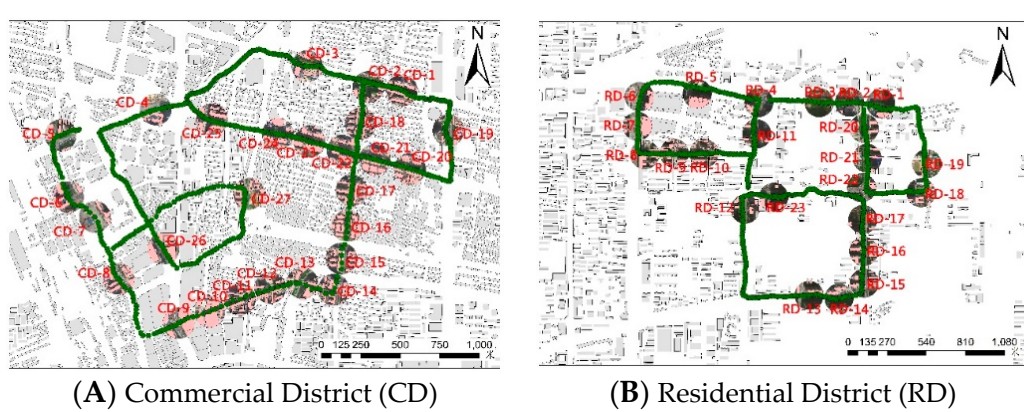

(**A**) Commercial District (CD)          (**B**) Residential District (RD)

**Figure 3.** Locations of the Er-qi commercial district and 27 sampling circle areas (**A**), locations of the Yanzhuang residential district and 23 sampling circle areas (**B**).

**Table 1.** Urban morphology index of each site of the commercial district.

| Sites | Plot Ratio (PR) | Building Density (BD %) | Building Height (BH m) | Green Coverage Rate (GCR %) |
|---|---|---|---|---|
| CD1 | 2.10 | 33.83 | 17.3 | 20.27 |
| CD2 | 1.00 | 15.18 | 16.3 | 41.81 |
| CD3 | 1.73 | 35.8 | 12.4 | 31.5 |
| CD4 | 1.02 | 16.85 | 13.5 | 0.72 |
| CD5 | 0.56 | 21.7 | 7.6 | 19.38 |
| CD6 | 1.92 | 24.58 | 16.5 | 5.12 |
| CD7 | 0.90 | 10.04 | 21.6 | 2.43 |
| CD8 | 2.15 | 29.66 | 22.8 | 5.71 |
| CD9 | 2.03 | 46.07 | 12.4 | 14.69 |
| CD10 | 1.58 | 48.46 | 12.3 | 13.99 |
| CD11 | 1.46 | 24.71 | 18.3 | 15.11 |
| CD12 | 2.00 | 29.46 | 17.8 | 8.15 |
| CD13 | 1.40 | 29.13 | 12.9 | 9.87 |
| CD14 | 1.74 | 34.9 | 13.9 | 7.57 |
| CD15 | 1.68 | 28.75 | 18.2 | 6.95 |
| CD16 | 2.00 | 46.65 | 14.1 | 13.89 |
| CD17 | 2.41 | 31.75 | 18.9 | 15.81 |
| CD18 | 1.82 | 28.24 | 16.4 | 20.6 |
| CD19 | 3.13 | 24.39 | 19.8 | 25.73 |
| CD20 | 2.46 | 38.07 | 17.7 | 16.08 |
| CD21 | 1.71 | 34.98 | 16.3 | 14.07 |
| CD22 | 3.08 | 24.35 | 27.8 | 10.95 |
| CD23 | 3.19 | 37.22 | 19.9 | 17.51 |
| CD24 | 3.70 | 39.78 | 19.1 | 15.59 |
| CD25 | 3.98 | 33.17 | 28 | 23.67 |
| CD26 | 2.03 | 60.18 | 10.4 | 8.12 |
| CD27 | 1.21 | 27.7 | 11.5 | 26.63 |

In this study, we selected the Yanzhuang residential district to represent typical residential districts in Zhengzhou. This is a newly built high-rise residential district, with mostly 15-story high-rise residential buildings. The large-area, high-rise, low-density residential structure represented by this district is unique to China. Its spatial form differs from that of a traditional residential district in that the sizes of the buildings are relatively large, and their influence on the surrounding environment is significant. Thus, this district has relatively high research value.

The circumference of the district is approximately 6.6 km, and the roads are 8-lane arterials that are approximately 27 m wide. We selected 23 circular sampling sites, each with a radius of 100 m, for this study; these 23 sampling circles' BD range from 0.35 to 29.49%, with an average of 14.76%, PR range from 0 to 2.83, with an average of 1.43, BH range from 6 m to 76 m, with an average of 32.29 m, and GCR range from 3.33 to 56.48%, with an average of 19.12%; their locations are shown in Figure 3B. The mean specific indexes of urban spatial form for each sampling circle are shown in Table 2.

**Table 2.** Urban morphology index of each site of the residential district.

| Sites | Plot Ratio (PR) | Building Density (BD %) | Building Height (BH m) | Green Coverage Rate (GCR %) |
|---|---|---|---|---|
| RD1 | 1.12 | 5.47 | 42.75 | 31.42 |
| RD2 | 0.37 | 5.38 | 23.10 | 25.75 |
| RD3 | 0.00 | 0.35 | 7.50 | 56.48 |
| RD4 | 0.52 | 4.69 | 33.00 | 17.05 |
| RD5 | 2.67 | 28.52 | 31.00 | 21.20 |
| RD6 | 1.10 | 20.21 | 48.00 | 3.33 |
| RD7 | 2.83 | 29.49 | 38.50 | 11.57 |
| RD8 | 1.61 | 21.08 | 21.50 | 7.01 |
| RD9 | 1.62 | 26.41 | 15.64 | 8.74 |
| RD10 | 1.68 | 26.81 | 12.60 | 5.55 |
| RD11 | 1.21 | 11.47 | 35.40 | 14.14 |
| RD12 | 1.04 | 7.06 | 41.25 | 11.06 |
| RD13 | 2.42 | 10.57 | 76.00 | 22.98 |
| RD14 | 2.09 | 15.67 | 38.25 | 20.61 |
| RD15 | 1.27 | 14.41 | 26.63 | 13.86 |
| RD16 | 2.52 | 25.56 | 28.64 | 23.74 |
| RD17 | 2.27 | 10.67 | 60.75 | 19.94 |
| RD18 | 1.73 | 14.78 | 28.50 | 15.52 |
| RD19 | 0.07 | 3.40 | 6.00 | 17.01 |
| RD20 | 1.05 | 16.55 | 28.50 | 21.17 |
| RD21 | 1.52 | 12.93 | 22.88 | 19.79 |
| RD22 | 0.67 | 11.33 | 37.50 | 20.77 |
| RD23 | 1.53 | 16.23 | 21.43 | 9.42 |

*2.2. Methodology and Settings*

In this study, we used bicycles to collect mobile temperature and humidity data within each sample site. We established three travel routes within the study commercial and residential districts and simultaneously collected data along each route; the routes are shown in Figure 3. These routes were established to ensure that each surveyor's workload was manageable. Surveyors all began collecting measurements simultaneously along the three routes. A temperature meter and moisture meter were carried on the bike baskets as shown in Figure 4B; a zip-top can wrapped with aluminum foil was used as a simple shutter box to prevent the influence of solar radiation on the instrument. Measurements began on 22 July 2017, and continued until 25 July 2017. Measurements were taken between 5:30–6:00 a.m., during the period sunrise and when the temperature starts to rise. The hottest period in the urban area is 1:30–2:00 p.m. Measurements were also taken at 9:15–9:45 p.m., which is the period in which the sun has set and the temperature starts to decrease. Due to the influence of traffic conditions at different times of the day, the length of each survey varied slightly but generally concluded within approximately 30 min. According to the data

published by the Zhengzhou City Meteorological Bureau, the weather from 22 July 2017 to 25 July 2017 was, respectively, cloudy to sunny and overcast to cloudy, and air temperatures ranged from 25–38 °C. Grade-3 and grade 3–4 winds predominated, and some change in the dominant wind direction occurred each day. The details are shown in Table 3.

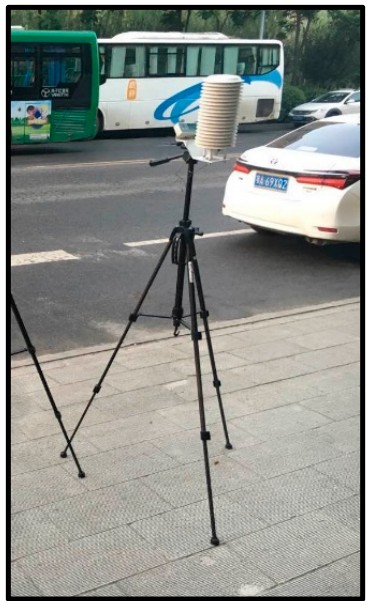

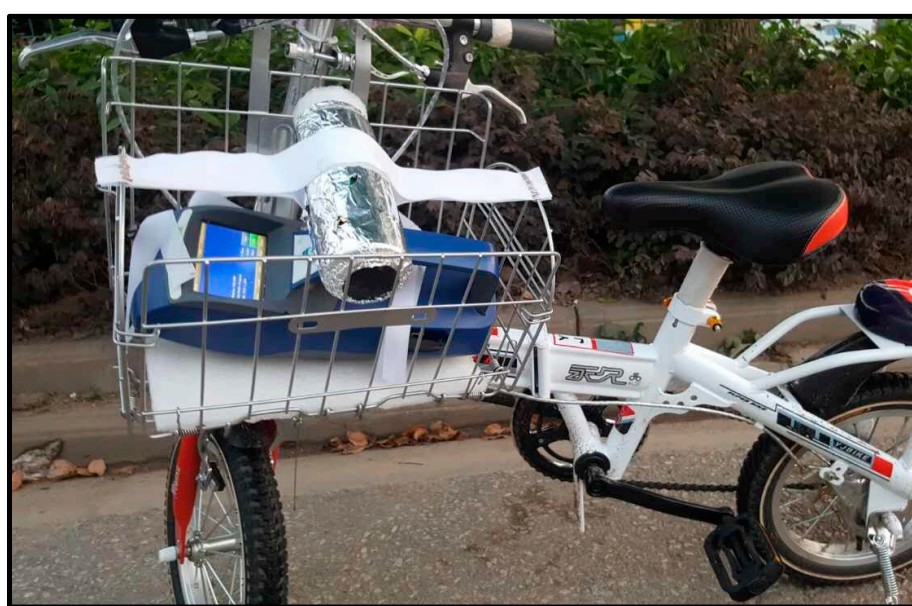

(**A**) Reference Point  (**B**) Mobile measurement

**Figure 4.** Instruments' installations of the reference point (**A**) and for mobile measurements (**B**).

**Table 3.** Meteorological data of the measurement days.

| Date | Max. Air Temperature °C | Min. Air Temperature °C | Weather | Wind Direction | Wind Level |
|---|---|---|---|---|---|
| 22/7/2017 | 36 | 28 | cloudy/sunny | east/east south | gentle to moderate |
| 23/7/2017 | 37 | 28 | cloudy/sunny | south | less than gentle |
| 24/7/2017 | 38 | 28 | overcast/cloudy | east north | gentle to moderate |
| 25/7/2017 | 33 | 25 | overcast/cloudy | north/east | less than gentle |

All mobile measurements in this study were conducted using bicycles. Each bike had a basket in the front, which contained a motion camera, temperature and humidity self-recording instruments, and a GPS. Data were obtained by travelling slowly along the route at a constant speed (10 km/h); street view video, temperature, humidity, and GPS tracking data were collected. Simultaneously, we collected data from a fixed point to establish background temperature and humidity characteristics for reference and data processing. The fixed point for this survey was established beside the Foreign Language Primary School on Cultural Palace Road in Zhengzhou (113.622151° E, 34.744031° N), as shown in Figure 1. Dedicated staff also obtained thermal images of each sampling area during the study period to better inform our understanding of the heat sources in cities and where they are concentrated. Instrument equipment types and settings used during the study are shown in Table 4.

**Table 4.** Details of the measurement instruments.

| Instruments | Purpose | Product Description | Measurement Interval | Picture |
|---|---|---|---|---|
| Tripod | Retainer | —— | —— | —— |
| Screen | Instrument shelter | —— | —— | —— |
| Thermal infrared camera | Record surface temperature | FLUKE Ti450 | —— |  |
| Temperature/humidity data logger | Temperature and relative humidity | YOWEXA SSN-22 precision accuracy: 0.1 °C, 0.1% deviation: ±0.2 °C, ±1% range: −35~80 °C, 0~100% | 10 s |  |
| GPS | Record the measured track | GARMIN Etrex 201x precision accuracy: 2.2 m | 10 s |  |
| GoPro | Record the process of mobile measurement | IDV SQ9 | —— |  |

## 3. Results

Figure 5 shows the four-day average temperature and humidity at each reference point along the six mobile measurement routes and at the stationary reference point. ZC-1, ZC-2, and ZC-3 are three routes of commercial districts; ZD-1, ZD-2, and ZD-3 are three routes of residential districts. Generally, the air temperature within the commercial district was higher than that in the residential district and at the reference point. Only when the solar elevation was relatively high in the afternoon did air temperatures in both districts become similar, while air temperatures in the residential district were relatively low in the morning and in the evening. These data suggest that, given the relatively low altitudinal angle of sun in the morning, the high-rise residences throughout the district had a shading effect on the surrounding streets. These structures obscure most of the direct sunlight, causing relatively low street temperatures. After sunset, temperature decreases in the residential district were significant; however, no obvious decreases were seen in the commercial district. This suggests that the relatively high artificial heat emissions associated with commercial activities have an impact on temperatures within the commercial district. The difference between the average humidity in the commercial and residential districts was relatively

small. Only in the morning were humidity values in the residential district obviously greater than those in the commercial district, which is likely related to the shade effect of the high-rise buildings. Because the reduction in direct sunlight causes a reduction in evaporation, relative humidity increases.

Figures 6 and 7 show the distribution of the four-day average temperature and humidity ratios for the mobile measurements in the commercial district, residential district, and at the reference point. Air temperature ratio and humidity ratio refer to air temperatures and humidity for the points of the measurement routes divided by air temperatures and humidity of reference points. These results clearly indicate the relationship between high temperatures, building density, and building volume in different land parcels within a high humidity region and its surrounding areas.

Relatively high commercial district temperature ratios were concentrated in the west and southwest. Building volumes are relatively large in this region, and, therefore, artificial heat emissions are also higher than in other regions. Furthermore, while building densities are relatively high in the east and northeast of the study area, the occlusion between buildings is relatively small in the western and southwestern areas, where the temperature ratio is higher.

The temperature and humidity ratio in the high-rise, low-density residential district varies slightly from that of the commercial district, as shown in Figure 7. Because building heights in this district are relatively large, high temperature ratio areas were primarily concentrated where the shading effect of buildings is not obvious, and building density was relatively low. The road surface in high temperature ratio areas was exposed to direct sunlight, especially near the arterial roads. The intense heat emissions from traffic resulted in relatively high temperature ratios in the morning and evening.

To explore whether the driving mechanisms of air temperature and relative humidity differences varied with times of day, we further analyzed the relationship of air temperature and relative humidity between morning and afternoon, morning and night, and afternoon and night. Pearson's correlation analysis was done for both of the cases, and Pearson's correlation coefficients between air temperatures (relative humidity) at different times of day for the commercial district and the residential district are given in Tables 5–8, respectively. In these two tables, MT, AT, NT, MH, AH, and NH represent morning air temperature, afternoon air temperature, night air temperature, morning relative humidity, afternoon relative humidity, and night relative humidity, respectively. Max, Min, and Ave give the maximum, minimum, and average values of air temperature (relative humidity) at different times of day.

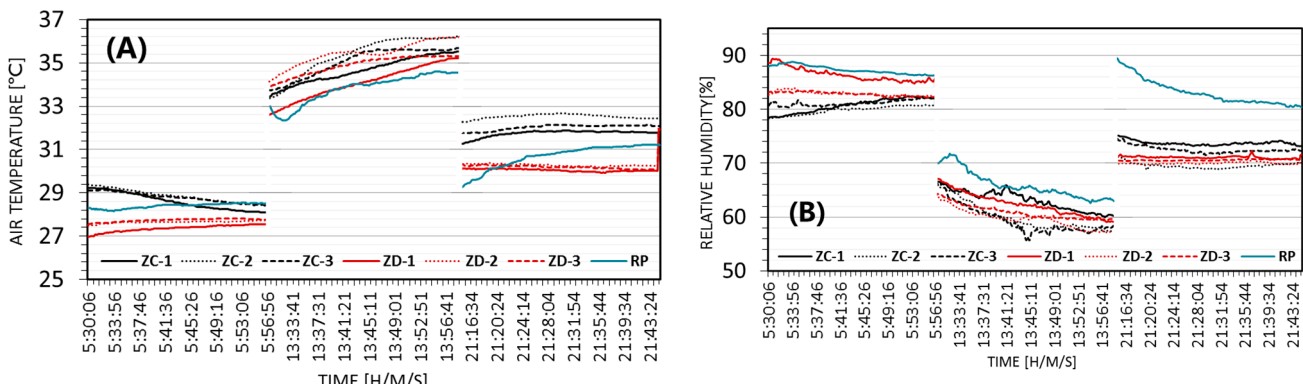

**Figure 5.** Four-day average (**A**) air temperatures and (**B**) relative humidity for six of the mobile measurement routes and for the reference point.

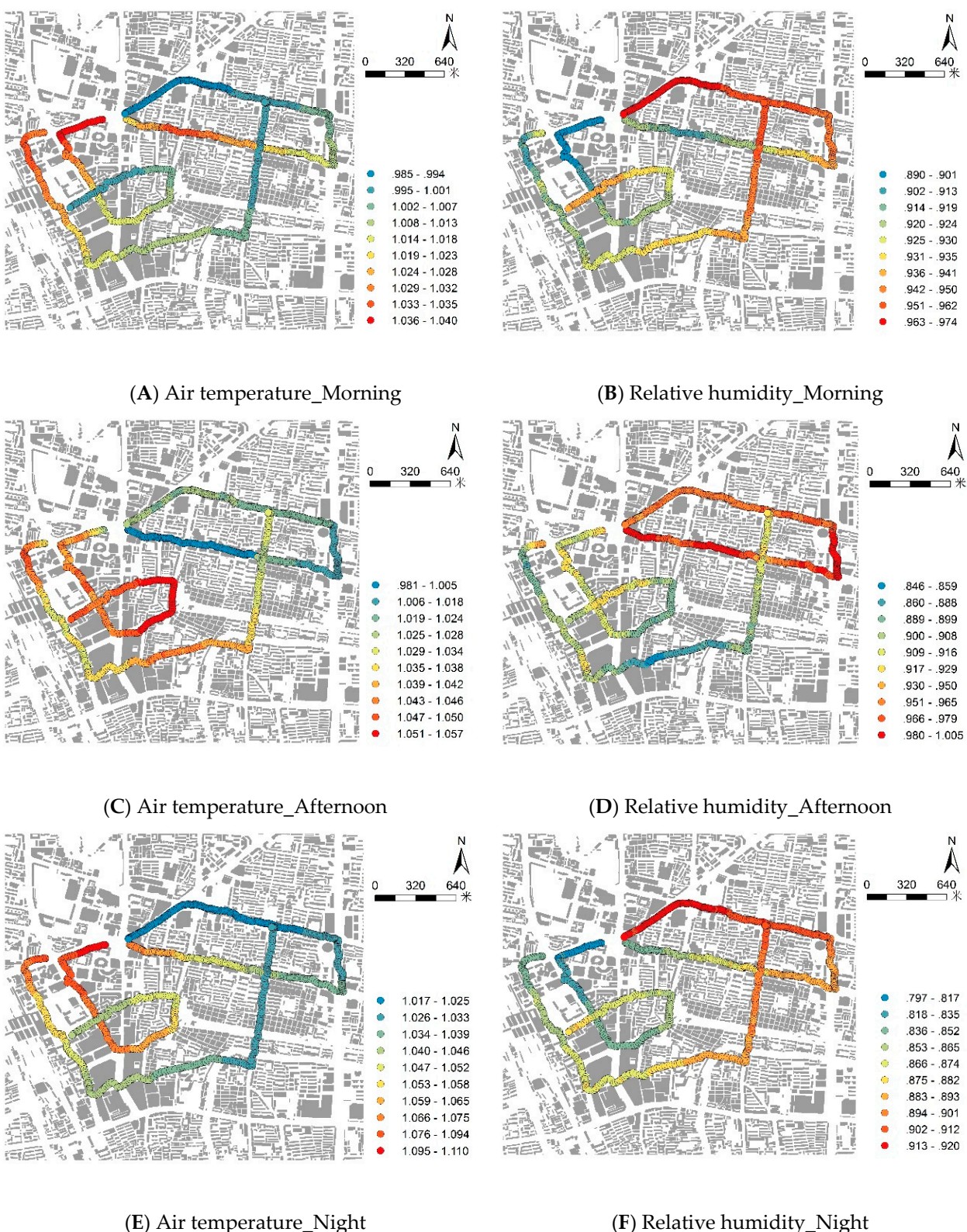

**Figure 6.** Distributions of four-day average air temperature ratio and relative humidity ratio of the Er-qi commercial district.

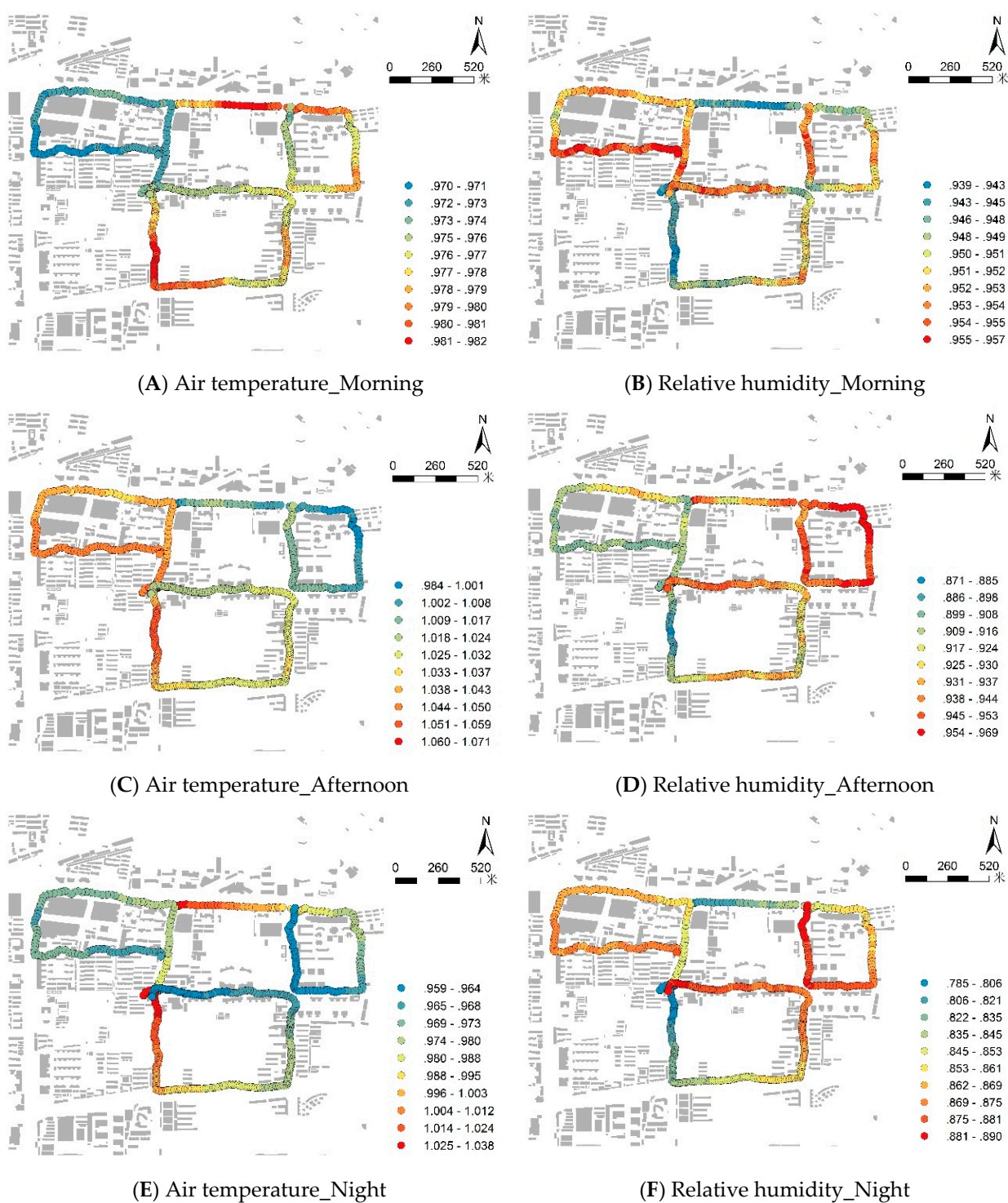

**Figure 7.** Distributions of four-day average air temperature ratio and relative humidity ratio of the Yanzhuang residential district.

**Table 5.** Pearson's correlation coefficients between air temperatures (relative humidity) at different times of the day for the commercial district (*n* = 27).

| AT | MHMax | MHMin | MHAve | AHMax | AHMin | AHAve | NHMax | NHMin | NHAve | RH |
|---|---|---|---|---|---|---|---|---|---|---|
| MTMax | 1 | 0.849 ** | −0.008 | −0.480 * | −0.625 ** | −0.540 ** | 0.237 | 0.184 | 0.220 | MHMax |
| MTMin | 0.955 ** | 1 | 0.033 | −0.717 ** | −0.665 ** | −0.723 ** | 0.056 | 0.122 | 0.088 | MHMin |
| MTAve | 0.993 ** | 0.981 ** | 1 | −0.170 | −0.133 | −171 | −0.228 | −0.195 | −0.210 | MHAve |
| ATMax | −0.386 * | −0.442 * | −0.411 * | 1 | 0.897 ** | 0.989 ** | 0.041 | −0.063 | −0.011 | AHMax |
| ATMin | −0.472 * | −0.414 * | −0.456 * | 0.892 ** | 1 | 0.948 ** | −0.098 | −0.113 | −0.119 | AHMin |
| ATAve | −0.460 * | −0.434 * | −0.456 * | 0.937 ** | 0.993 ** | 1 | 0.004 | −0.075 | −0.039 | AHAve |
| NTMax | 0.507 ** | 0.499 ** | 0.505 ** | −0.015 | −0.007 | −0.002 | 1 | 0.970 ** | 0.994 ** | NHMax |
| NTMin | 0.559 ** | 0.489 ** | 0.535 ** | −0.025 | 0.072 | 0.054 | 0.976 ** | 1 | 0.989 ** | NHMin |
| NTAve | 0.532 ** | 0.483 * | 0.515 ** | −0.034 | 0.032 | 0.022 | 0.990 ** | 0.997 ** | 1 | NHAve |
|  | MTMax | MTMin | MTAve | ATMax | ATMin | ATAve | NTMax | NTMin | NTAve | |

\* Correlation is significant at an $\alpha$ = 0.05 level (two-tailed); ** Correlation is significant at an $\alpha$ = 0.01 level (two-tailed).

**Table 6.** Pearson's correlation coefficients between air temperature ratio (relative humidity ratio) and urban morphology index for the commercial district (*n* = 27).

|  |  | BD (%) | PR | BH (m) | GCR (%) |
|---|---|---|---|---|---|
| Average Air Temperature Ratio | Morning | −0.181 | 0.156 | 0.227 | −0.508 ** |
|  | Afternoon | 0.006 | −0.772 ** | −0.646 ** | −0.229 |
|  | Night | −0.148 | −0.114 | −0.090 | −0.427 * |
| Average relative humidity ratio | Morning | 0.060 | −0.086 | −0.087 | 0.589 ** |
|  | Afternoon | −0.014 | 0.642 ** | 0.452 * | 0.409 * |
|  | Night | 0.079 | 0.089 | 0.088 | 0.471 * |

\* Correlation is significant at an $\alpha$ = 0.05 level (two-tailed). ** Correlation is significant at an $\alpha$ = 0.01 level (two-tailed).

**Table 7.** Pearson's correlation coefficients between air temperatures (relative humidity) at different times of day for the residential district (*n* = 23).

| AT | MHMax | MHMin | MHAve | AHMax | AHMin | AHAve | NHMax | NHMin | NHAve | RH |
|---|---|---|---|---|---|---|---|---|---|---|
| MTMax | 1 | 0.793 ** | 0.867 ** | 0.743 ** | 0.692 ** | 0.548 ** | 0.335 | 0.155 | 0.244 | MHMax |
| MTMin | 0.720 ** | 1 | 0.979 ** | 0.456 * | 0.777 ** | 0.560 ** | 0.234 | 0.258 | 0.258 | MHMin |
| MTAve | 0.966 ** | 0.819 ** | 1 | 0.528 ** | 0.787 ** | 0.578 ** | 0.257 | 0.221 | 0.249 | MHAve |
| ATMax | 0.764 ** | 0.594 ** | 0.783 ** | 1 | 0.746 ** | 0.705 ** | 0.353 | 0.069 | 0.217 | AHMax |
| ATMin | 0.548 ** | 0.845 ** | 0.656 ** | 0.761 ** | 1 | 0.741 ** | 0.189 | 0.178 | 0.202 | AHMin |
| ATAve | 0.709 ** | 0.641 ** | 0.775 ** | 0.967 ** | 0.843 ** | 1 | 0.762 ** | 0.674 ** | 0.753 ** | AHAve |
| NTMax | 0.227 | −0.175 | 0.116 | 0.0015 | −0.373 | −0.137 | 1 | 0.873 ** | 0.965 ** | NHMax |
| NTMin | 0.251 | 0.472 * | 0.367 | 0.156 | 0.391 | 0.270 | 0.225 | 1 | 0.968 ** | NHMin |
| NTAve | 0.324 | 0.166 | 0.300 | 0.203 | 0.093 | 0.178 | 0.382 | 0.247 | 1 | NHAve |
|  | MTMax | MTMin | MTAve | ATMax | ATMin | ATAve | NTMax | NTMin | NTAve | |

\* Correlation is significant at an $\alpha$ = 0.05 level (two-tailed); ** Correlation is significant at an $\alpha$ = 0.01 level (two-tailed).

**Table 8.** Pearson's correlation coefficients between air temperature ratio (relative humidity ratio) and urban morphology index for the residential district (*n* = 23).

|  |  | BD (%) | PR | BH (m) | GCR (%) |
|---|---|---|---|---|---|
| Average Air Temperature Ratio | Morning | −0.682 ** | −0.353 | 0.037 | 0.774 ** |
|  | Afternoon | 0.700 ** | 0.546 ** | 0.191 | −0.539 ** |
|  | Night | −0.378 | −0.380 | 0.101 | 0.413 |
| Average relative humidity ratio | Morning | 0.652 ** | 0.470 * | −0.109 | −0.708 ** |
|  | Afternoon | −.652 ** | −0.450 * | −0.020 | 0.437 * |
|  | Night | 0.579 ** | 0.424 * | −0.088 | −0.277 |

\* Correlation is significant at an $\alpha$ = 0.05 level (two-tailed); ** Correlation is significant at an $\alpha$ = 0.01 level (two-tailed).

There was a moderate negative correlation between the air temperatures in the morning and afternoon for the commercial district ($r$ values from $-0.472$ to $-0.386$). In comparison, the absolute values of the correlation coefficients between the air temperatures in the morning and night were quite similar ($r$ values from $0.483$ to $0.559$), but the correlation was just the opposite. The negative relationships between air temperature in the morning and afternoon indicate that key factors that caused this commercial district's air temperature to rise were quite different at different times of day. To find the key factors of the commercial district warming, we further analyzed the correlation between air temperature ratio (relative humidity ratio) and urban morphology index in the morning, afternoon, and night (results are shown in Table 6). During morning and night, air temperature ratio (to eliminate the effect of diurnal temperature rise) had a moderate negative correlation with GCR ($r = -0.508$ morning; $r = -0.427$ night) only, whereas, in the afternoon, air temperature ratio was related to PR ($r = -0.772$) and BH ($r = -0.646$). According to the results from Tables 5 and 6, we presume that the negative relationship between air temperatures in the morning and afternoon was mainly due to the warming mechanism differences. Urban morphology indexes related to the air temperature ratio were selected, and orders and trends between the data are presented in Figure 8. As shown, during the morning and night, GCR is the key factor for cooling the atmosphere, and, therefore, the greater the greenery areas in the commercial district, the cooler that district will be. Furthermore, the two fitting lines representing morning and night share almost the same slope, which illustrates that the cooling effects of vegetation in the morning and night are identical with each other. However, in the afternoon, PR and BH are the key factors, and, moreover, they have a strong negative correlation with air temperature ratio, which demonstrates that an open space without shelter from surrounding buildings in that district is highly possible to be hotter than the other areas in the summer. This explains why the air temperatures in the morning and afternoon had a negative relationship, as known, and usually a built-up area leads to low GCR.

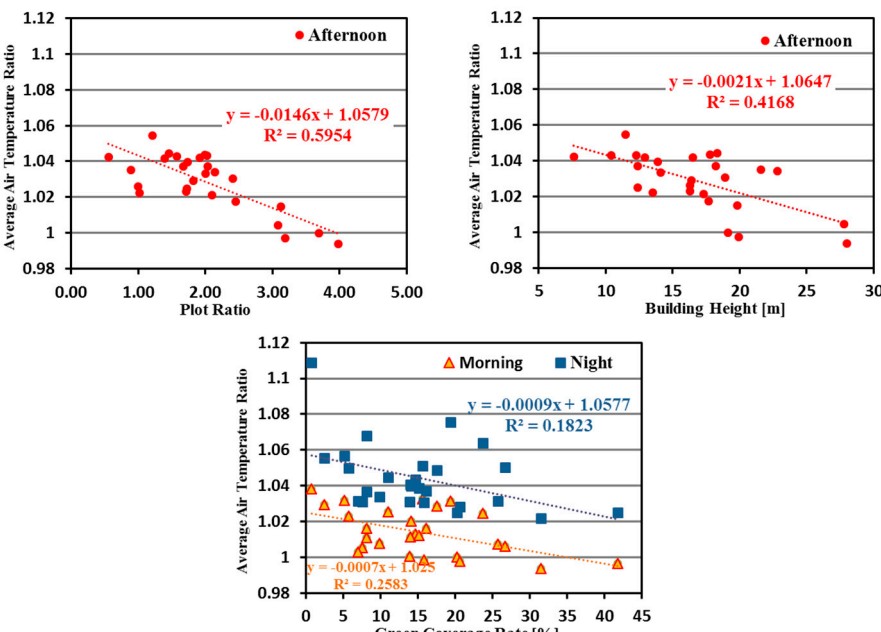

**Figure 8.** Relationships between Urban morphology index (*x*-axis) and air temperature ratio from the mobile and reference point (*y*-axis) measurements in the commercial district.

A negative relationship can also be found between morning relative humidity (max. and min.) and afternoon relative humidity ($r$ values from $-0.48$ to $-0.723$) in Table 5, similar to the air temperature results. The reason for this negative result remains the same, which is that the key factors influencing the atmosphere were distinct at different times of the day. As shown in Table 6, GCR, which affects relative humidity by transpiration

and evaporation, is the core index related to relative humidity in the morning (*r* = 0.589), afternoon (*r* = 0.409), and night (*r* = 0.471). In addition to GCR, PR and BH also had strong (*r* = 0.642) and moderate (*r* = 0.452) correlations with relative humidity in the afternoon. These results illustrate that, when comparing built-up areas with open space in this commercial district, built-up areas with higher PR and BH might have higher relative humidity, and we assumed that is just due to their lower air temperature.

Urban morphology indexes that were related to the relative humidity ratio were selected, and orders and trends between the data are presented in Figure 9. As shown, relative humidity ratio is increasing while PR, BH, and GCR increase in the afternoon in this commercial district, whereas, in the morning and night, only GCR had an impact on that. Nevertheless, the slope of the fitting line for GCR in the afternoon is the highest, while the slopes for the morning and night are almost the same. Why is the humidification efficiency in the afternoon the highest? We speculated that the higher solar radiation of that moment than for the other two periods was the primary cause of this phenomenon.

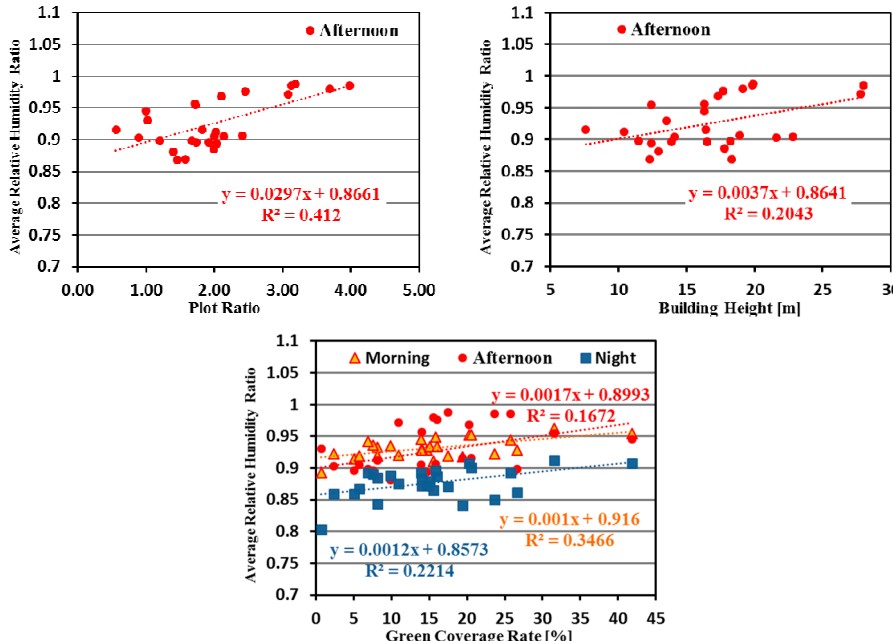

**Figure 9.** Relationships between urban morphology index (*x*-axis) and relative humidity ratio from the mobile and reference point (*y*-axis) measurements in the commercial district.

It was quite a different situation in the residential district, which can be seen from Tables 7 and 8. The relevance between air temperatures in the morning and afternoon was moderate to extremely strong (*r* values from 0.548 to 0.845). The reason for that is obvious in Table 8—the key factors that influenced air temperature in the morning and night were quite similar. However, the way these elements affected the thermal atmosphere in the morning and night was different. As shown in Table 8, BD had a negative strong correlation with air temperature ratio in the morning (*r* = −0.682) and a strong positive correlation with that in the afternoon (*r* = 0.700). GCR had a positive strong correlation with air temperature ratio in the morning (*r* = 0.774), while, in the afternoon, it had a moderate negative correlation (*r* = −0.539). Moreover, PR was simply related to air temperature ratio in the afternoon (*r* = 0.546).

On the basis of Table 8, part of the data were fitted and are given in Figure 10. The fitting line's slope is gentler in the morning than in the afternoon, which indicates that the thermal environment in the afternoon for this residential district is more affected by urban morphology than in the morning. Air temperature ratio was higher while BD was lower, and GCR was higher in the morning; on the contrary, during the afternoon, air temperature ratio was higher while BD was higher, and GCR was lower. We speculated that, due to the building shading effect, air temperature was lower in the area with building shelters in this

residential district in the morning—whereas, in the afternoon, particularly at approximately 1:00 p.m., solar elevation was highest in a day's period, building shading was not effective, and, moreover, an array of buildings blocked ventilation and heat dissipation. During night, air temperature had no connection with morning and afternoon air temperature, even with any index of urban morphology. Night air temperature in this residential district was almost 2 °C cooler than in the commercial district (shown in Figure 5), which illustrated that, for this residential district in this cold region city, heat was well dissipated during the night, and night air temperature was not affected by the surrounding built environment.

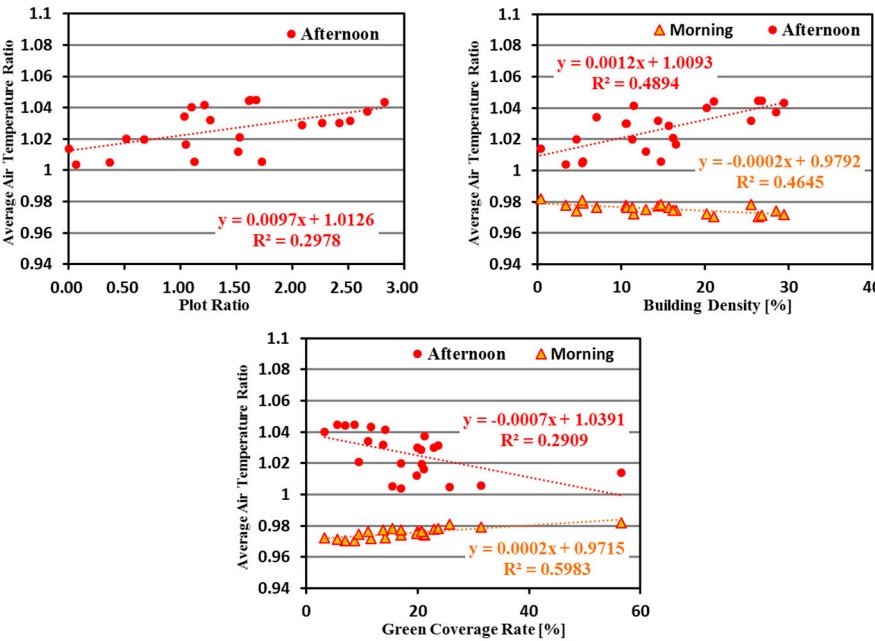

**Figure 10.** Relationships between urban morphology index (*x*-axis) and air temperature ratio from the mobile and reference point (*y*-axis) measurements in the residential district.

According to Figure 5, the commercial district's air temperature was 1 to 2 °C higher than that in the residential district in the morning and night. The reasons for the tremendous temperature differences between these two districts are probably that the average BD for the commercial district is as much as twice that for the residential district (shown in Tables 1 and 2) and the large amount of anthropogenic heat release due to flourishing economic activities in the commercial district. Because BD in the residential district is much lower, and in the early afternoon, solar elevation is almost 90°, building shelters will not effectively prevent heat accumulation in street canyons. In the commercial district, however, cooling the atmosphere mainly relies on the building shading effect.

As for relative humidity in the residential district, there were moderate to strong correlations between morning and afternoon (*r* values from 0.456 to 0.787), and average relative humidity in the afternoon had a strong relationship with that at night (*r* values from 0.674 to 0.762). Relative humidity ratio in the morning had strong ($r = 0.652$), moderate ($r = 0.470$), and strong negative ($r = -0.708$) correlations with BD, PR, and GCR, respectively. In the afternoon, it had strong negative ($r = -0.652$), moderate negative ($r = -0.450$), and moderate ($r = 0.437$) correlations with BD, PR, and GCR, respectively. During the night, relative humidity ratio had moderate relationships both with BD ($r = 0.579$) and PR ($r = 0.424$). It is widely known that relative humidity has an inverse correlation with air temperature. Therefore, in the morning and afternoon, the correlation of relative humidity ratio and urban morphology indexes is contrary to that of air temperature ratio and urban morphology indexes. During the night, air temperature ratio had no relationship with any of the morphology indexes, but relative humidity ratio increased while BD and PR

increased. We presumed that relative humidity increases, whereas, in a closuring space, evaporation capacity decreases.

On the basis of Table 8, part of the data were selected and fitted; they are shown in Figure 11. From that figure, it is not difficult to find that the relative humidity ratio is more sensitive to urban morphology changes at night than in the other periods. However, no analogous results have been found in the commercial district. Traced to its cause, presumably this is because relative humidity in the commercial district is influenced by multiple factors, for instance, exhaust humidity from air conditioners of gigantic commercial buildings, metabolism of the crowded people, transportation, and large equipment in that area.

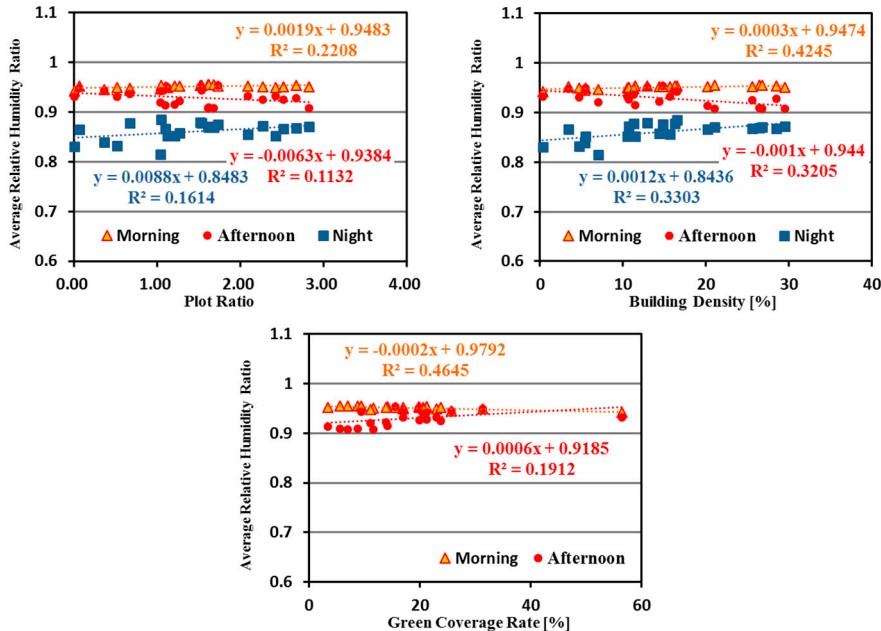

**Figure 11.** Relationships between the urban morphology index (*x*-axis) and relative humidity ratio from the mobile and reference point (*y*-axis) measurements in the commercial district.

## 4. Discussion

Through the mobile measurements in Zhengzhou in the summer, we learned that BD, PR, BH, and GCR had varying degrees of impacts on the air temperatures of commercial and residential districts. To different district classifications, the way that urban morphology may influence surrounding micro-climate is variant, and the effect is even discrepant at different times of the day. BD, for example, has no direct relevance with air temperature in the commercial district, where BD is comparably high, but it is strongly correlated with air temperature in the residential district. Moreover, the ways in which BD impacts air temperature in the morning and afternoon are opposite. Therefore, further discussion was initiated. The increase of BD may lead to larger shadow areas in street canyons and may therefore decrease the absorption of direct solar shortwave radiation and decrease air temperature in street canyons for low solar elevations. Nevertheless, when the solar elevation is high, building shading is not effective, more shortwave and longwave radiation is absorbed by multiple reflections due to high BD, and less exhaust heat is dissipated by convection due to the narrow street canyons. This phenomenon was also discovered in Yang and Li's study [30]. In addition, Guo et al. [31] characterized the impact of urban morphology heterogeneity on land surface temperature (LST) in Guangzhou and discovered that urban morphology of medium building height and lower density significantly yielded higher LST variation levels, whereas the lowest LST variation levels occurred in high-rise and high-dense building arrays. This is quite different from Zhengzhou's results. From our point of view, the difference is mainly due to the shading effect in cold and hot regions of China. By comparing the results of the present study to the Wuhan measurements, we

found that, given the summer high temperature in Zhengzhou, a cold-climate city, the air temperature is significantly influenced by direct sunlight, as mediated by the shading effect. Wuhan is located in a region with hot summers and cold winters; there, the shading effect does not have a large influence on the rise and fall of air temperature [32]. However, there is still some common law in both cold region and hot regions, that is, compared to building height, building density had a stronger influence on LST [31].

It is worth mentioning that the influence of urban morphology on air temperature during Zhengzhou's night was not obvious; from our point of view, exhaust heat was well dissipated during the night, and air temperature in this northern Chinese cold region city was low. However, Huang et al. [33] measured Nanjing, which is a hot summer and cold winter city in China; they found that, in Nanjing, air temperature was still high due to its weak cooling effect during the night, especially in a fully built area. Moreover, when residents are suffering through a hot night, they may rely on air conditioners more. As a vicious circle, even larger amounts of anthropogenic heat exhaust are released to street canyons, making them even hotter. Moreover, as mentioned above, a higher UHI intensity in the southeastern parts of China during the day can be detected, and UHI intensity in the northern regions of the country is higher at night compared with other regions. In particular, the northeast, with a humid-cold climate, experienced the most intense UHI intensity across China [7]. Inspired by this phenomenon, aiming at different climatic regions, different urban morphologies should be adopted. For instance, in cold region cities, relatively high BD and BH are acceptable for a comparably comfortable thermal environment in the summer, but, in hot region cities, rigorous control of BD and BH for shading effect during the daytime and efficient convection at night is needed to create a cooler environment.

In addition to urban morphology impacts, anthropogenic heat impacts also attracted our attention during this mobile measurement project. It is widely known that urban micro-climates are dominated by local climate and urban morphology, and, in addition, residents' activities may form disparate micro-climates even in the same city by releasing anthropogenic heat. Through this measurement project, we found that heat released from the equipment of giant volume commercial buildings and endless streams of traffic played dominant roles in commercial district anthropogenic heat release, whereas, in the residential district, air conditioner heat release was the primary source (shown in Figure 12). On the basis of the results, we knew that air temperature in the Er-qi commercial district was on average $1-2\,°C$ higher than that in the Yanzhuang residential district during the morning and night. We speculated that the higher air temperatures in the commercial district were partly due to the higher anthropogenic heat release. A similar study by Cao et al. [34] in Guangzhou revealed that, in different types of districts, anthropogenic heat release would cause air temperatures to rise differently. Air temperatures increased $0.32\,°C$, $0.19\,°C$, and $0.18\,°C$ in the commercial, high density residential, and low density residential districts of Guangzhou, respectively. Furthermore, Ma et al. [35] found that, during winter nights, $1.5\,°C$, $1.1\,°C$, and $0.7\,°C$ of warming occurred due to anthropogenic heat in high, medium, and low intensity regions, respectively. This warming effect declined to $<0.5\,°C$ during summer nights and further declined to $0.3\,°C$ during the daytime summer. Hamilton et al. [36] provided an explanation for this phenomenon. Short-wave radiation is dominant during the daytime, and anthropogenic heat release is much less than shortwave radiation; hence, the impact is quite limited during the daytime. However, during the night, when shortwave radiation vanished, longwave radiation is dominant, the amount of which is as great as anthropogenic heat, and, thus, air temperature significantly increases with anthropogenic heat released during the night. Wang et al. [37] expressed anxiety about air temperature increases caused by anthropogenic heat. They believed that warming might result in a higher intensity Urban Heat Island phenomenon and even cause higher intensity urban local circulation, which might induce harmful effects on air pollutant diffusion.

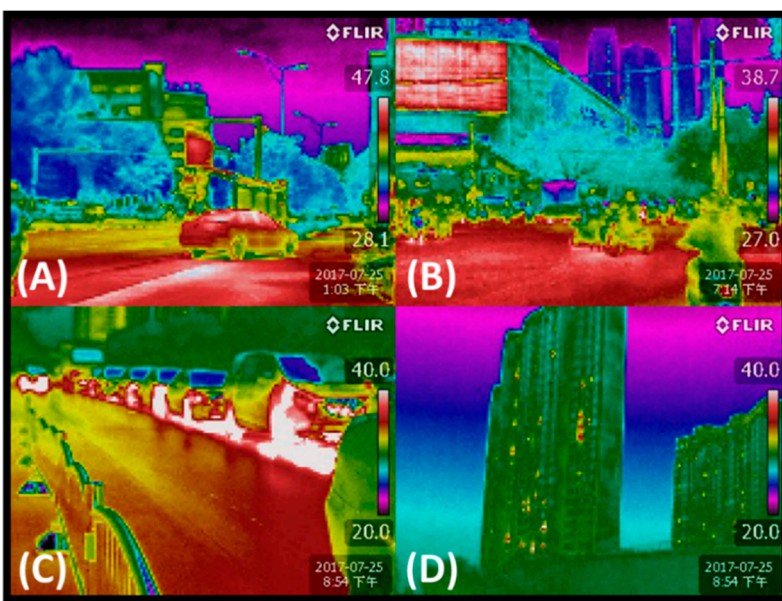

**Figure 12.** Infrared thermography of the Er-qi commercial district (**A**,**B**) and the Yanzhuang residential district (**C**,**D**).

Greenery areas having distinct impacts on micro-climate were noticed during this mobile measurement project. In the Er-qi commercial district, GCR increased and caused the air temperature ratio to decrease significantly during the morning and night. While in the Yanzhuang residential district, the cooling effect of greenery areas was mainly at work at approximately 1:00 p.m.; however, during the night, there was no clear correlation between GCR and air temperature ratio. Moreover, at approximately 5:00 a.m., there was a strong positive correlation between GCR and air temperature ratio, although, from the fitting line, we knew that air temperature just rose fractionally as GCR increased; nevertheless, it still meant something. We are not alone in this regard, as Giridharan et al. [38] noticed that air temperature is 0.5 to 1 °C cooler in greenery than in artificial underlying areas, which they conjectured is due to a lower reflection ratio in the artificial underlying area; hence, that area absorbed more heat flux and became increasingly warm. However, this cooling effect only seemed to happen in the areas with hot atmospheres. For example, at approximately 5:00 a.m. in the Yanzhuang residential district, when its average air temperature was approximately 28 °C, cooling effects of greenery areas were not obvious, and some of the areas even warmed its atmosphere. This phenomenon was mentioned in the study of Shuko and Takeshi [39]; vegetation is similar to a water body, and its cooling effect is at work when the surrounding area is hot, but, if the surrounding air temperature is low, greenery areas may have no impact on micro-climate or even the warming effect on the surrounding atmosphere. Therefore, they discovered that the diurnal air temperature difference was quite small in and around greenery areas. It is not a unique instance; Lai et al. [40] also mentioned in their study that vegetation plays different roles in regulating micro-climate in cold and hot regions. In hot regions, vegetation may offer more shading areas and cooling effects for the surrounding environment. However, in cold regions, vegetation can warm the atmosphere in the winter in addition to its cooling effect in the summer.

## 5. Summary and Conclusions

The present work indicates that the air temperature of the Er-qi commercial district was on average 1–2 °C higher than that of the Yanzhuang residential district during the morning and night. We speculate that the higher air temperature in the commercial district was partly due to the higher anthropogenic heat release. Moreover, micro-climates of the commercial and residential districts in this cold-region city were influenced by urban morphology quite differently.

In the Er-qi commercial district, the thermal micro-climate was related with PR and BH in the afternoon and with GCR during the morning and night. In the Yanzhuang residential district, however, the key urban morphology factors related to its thermal micro-climates were BD, PR, and GCR during almost the entire day. Especially in the afternoon, at approximately 1:00 p.m., air temperature ratio was significantly affected by urban morphology indexes; for instance, it had strong negative correlations with PR ($r = -0.772$) and BH ($r = -0.646$) in the commercial district, whereas, in the residential district, it had strong and moderate correlations with BD ($r = 0.700$), PR ($r = 0.546$) and GCR ($r = -0.539$). As we know, BD for the commercial district is two to three times that of the residential district, and BH for the commercial district is nearly 0.5 times that of the residential district. Therefore, high buildings everywhere in the residential district built natural shelters for street canyons at the pedestrian level, and increasing its BD may not offer extra effective shelters but rather obstacles, making exhaust heat difficult to dissipate. However, increasing BH in the commercial district can be quite effective in cooling that area by offering it much more shelter. It is worth mentioning that the influence of urban morphology on air temperature during Zhengzhou's nights was not obvious, and we found that exhaust heat was well dissipated during the night and that air temperature in this northern Chinese cold-region city was low.

We also observed that, in different situations, greenery areas can differently impact the thermal environments in the districts of this cold-region city. Vegetation and water bodies are similar in that their cooling effects are at work when the surrounding area is hot, but, if the surrounding air temperature is low, greenery areas may have no impact on micro-climate or even warming effect on the surrounding atmosphere. Therefore, in hot regions, vegetation may offer more shading areas and cooling effect to the surrounding environment. In cold regions, however, vegetation is able to warm the atmosphere in the winter in addition to its cooling effect in the summer.

This study has several limitations. The study did not specify an atmospheric stability situation during the observation dates. Atmospheric stability is a leading factor of heat trapping in the lowermost air layers. In future studies, we will firstly consider atmospheric stability. In addition, this is a short-term observation, in future study, we will consider collecting data over a long period to unveil the mechanism of urban morphology impact on urban climate.

**Author Contributions:** Conceptualization, X.Z. and H.C.; data curation, X.Z. Both authors have read and agreed to the published version of the manuscript.

**Funding:** This work was supported by the State Key Program of National Natural Science of China (Grant No. 51538004), the National Natural Science Fund Youth Project (Grant No. 51708237), the 57th China Postdoctoral Science Foundation Funded Project (Grant No. 2015M572144), the Independent Innovation Fund of Huazhong University of Science and Technology (Grant No. 0118220100), and the JURC Short-Term Internship Program.

**Institutional Review Board Statement:** The study was conducted according to the guidelines of the Declaration of Helsinki and approved by the Institutional Review Board.

**Informed Consent Statement:** Informed consent was obtained from all subjects involved in the study.

**Data Availability Statement:** Not applicable.

**Conflicts of Interest:** The authors declare no conflict of interest.

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
