# Peer review of "Experimental Analysis of the Influence of Urban Morphological Indices on the Urban Thermal Environment of Zhengzhou, China"

_atmosphere, doi:10.3390/atmos12081058_

Round 1

Reviewer 1 Report

Review of the manuscript “Analyzing the influence of urban morphological indexes on urban thermal environment of a cold-region city based on measurement” by Xuefan Zhou, Hong Chen

Major comments:

  1. I believe that introduction must be seriously modified. There is now a kind of cherry-picking assessment of the urban heat studies for hot climate cities. The paper is however targeting the cold-climate regions. There are many studies of cold climate cities, such as Edmonton, Montreal, Minneapolis, Helsinki, Moscow, Krasnoyarsk and many more, but they have not been used for the introduction. It creates a biased view in the paper with a lot of less relevant references and none references and findings of direct relevance to the subject.
  2. Data sources are poorly identified in the study. E.g., what dataset is used for Figure 2? Where the data come from? What do the data cover? Are the data accessible online? The same is true for the dataset sampled for the study itself. How a reader can access the data?
  3. The results of the study are given in absolute values only. It is not surprising that the sunlit sides of buildings have much higher temperature than the sides in shadow. So, to identify the impact of the BD, PR, BH, and GCR factors in the diurnal cycle, I would expect that the standard diurnal cycle is subtracted from the absolute values observed. Moreover, due to slower development of the atmospheric mixing, the efficiency of the heat convection in the atmosphere in the morning will be different from that in the evening. The maximum surface temperature and the maximum atmospheric temperature are shifted by 2-3 hours relative to each other. It needs to be taken into account when the diurnal cycle of impact is to analysis. So, I do not believe that a simple correlation analysis (from line 259 on), while it is probably correct, would tell us much about the acting factors.

Minor comments:

  1. Abstract: BD, PR, BH, and GCR are not yet defined
  2. Results: Figure 5 is hard to understand, please add empty space between separate time intervals and do not connect data between the intervals with lines.
  3. The study forgot to specify an atmospheric stability situation during the observation dates. Atmospheric stability is a leading factor of heat trapping in the lowermost air layers.

Author Response

Reviewer1:

Major comments:

  1. I believe that introduction must be seriously modified. There is now a kind of cherry-picking assessment of the urban heat studies for hot climate cities. The paper is however targeting the cold-climate regions. There are many studies of cold climate cities, such as Edmonton, Montreal, Minneapolis, Helsinki, Moscow, Krasnoyarsk and many more, but they have not been used for the introduction. It creates a biased view in the paper with a lot of less relevant references and none references and findings of direct relevance to the subject.

Response to the comment: Thank you for this comment, it is good suggestion. As you mentioned, we added several studies of cold climate cities in second last paragraph of introduction. As Zhengzhou city is located in the center of China, although it belongs to heating region of China. The average temperature of hottest and coldest months in Zhengzhou are 25.9℃(August) and 2.15℃(January). So strictly speaking, it is not an extremely cold region city, therefore we decide to revise the title of this article to “Analyzing the influence of urban morphological indexes on urban thermal environment of a cold-region city Zhengzhou, China based on measurement”.

2.Data sources are poorly identified in the study. E.g., what dataset is used for Figure 2? Where the data come from? What do the data cover? Are the data accessible online? The same is true for the dataset sampled for the study itself. How a reader can access the data?

Response to the comment: Thank you for this comment. Dataset source of Figure 2 is from http://www.weather.com.cn/forecast/history.shtml?areaid=101180101&month, supplied by China Meteorological Administration. And we have added this information in Figure2. Other data is mainly from surveys and measurement that we took in Zhengzhou city.

3.The results of the study are given in absolute values only. It is not surprising that the sunlit sides of buildings have much higher temperature than the sides in shadow. So, to identify the impact of the BD, PR, BH, and GCR factors in the diurnal cycle, I would expect that the standard diurnal cycle is subtracted from the absolute values observed. Moreover, due to slower development of the atmospheric mixing, the efficiency of the heat convection in the atmosphere in the morning will be different from that in the evening. The maximum surface temperature and the maximum atmospheric temperature are shifted by 2-3 hours relative to each other. It needs to be taken into account when the diurnal cycle of impact is to analysis. So, I do not believe that a simple correlation analysis (from line 259 on), while it is probably correct, would tell us much about the acting factors.

Response to the comment: This is a good question, and it is worth being well discussed. Actually the acting factors and working mechanism are dependent on urban canopy heat balance. As for heat gain, including longwave radiation, shortwave radiation, artificial heat emissions and so on. As for heat loss, mainly including latent heat, sensible heat, advection and heat storage. So why can the spatial morphology changes brought about by rapid urban expansion and development affect the urban microclimate? As for heat gain, first, a large amount of artificial heat emissions are generated in the work and life of urban residents. Beside that, the increase in building height leads to an increase in the urban skyline. The long-wave and short-wave radiation incident in the urban canopy are reflected multiple times between the building skins, thus increasing the total amount of absorption. For heat loss, as the natural underlying surface shrinks, the amount of latent heat loss brought by vegetation transpiration and water evaporation is reduced. Beside that, tall buildings obstruct air circulation, resulting in reduced advection, in other words, the potential for cooling through ventilation is reduced. Moreover, as the ratio of artificial underlay surface increases, heat storage materials in the urban canopy increase accordingly. This greatly improves the heat storage capacity of cities, and even causes high temperature night phenomenon.

Minor comments:

  1. Abstract: BD, PR, BH, and GCR are not yet defined

Response to the comment: Thank you for your kind suggestion. The definition of these abbreviations can be found from the last paragraph of introduction (building height (BH), building density (BD), green coverage rate (GCR), and plot ratio (PR)).

2.Results: Figure 5 is hard to understand, please add empty space between separate time intervals and do not connect data between the intervals with lines.

Response to the comment: Thank you for your kind suggestion. We have revised Fig. 5 according to the suggestion.  

3.The study forgot to specify an atmospheric stability situation during the observation dates. Atmospheric stability is a leading factor of heat trapping in the lowermost air layers.

Response to the comment: Yes, this is an important factor we missed in this study. Next time we won’t miss that when we do the experiment and measurement studies.

Reviewer 2 Report

Analyzing the influence of urban morphological indexes on urban thermal environment of a cold-region city based on measurement

This study tries to analyze the influence of urban morphological indexes on temperature and RH changes in the morning, afternoon, and night in residential and commercial properties in Zhengzhou, China. This city is located in cold regions and the interest is to find out if the urban morphological indexes act differently in cold vs hot regions.

Review questions:

  • In-text citations have different formats. Some include the first name, some are all in the capital, etc. Check and rewrite all the in-text citations.
  • There are some writing issues; you need to edit the draft. Make long sentences smaller so you can better explain matters.
  • Add the definition of Plot Ratio (PR) to the manuscript

Introduction

  • Line 31-33: The sentence is not fluent.
  • Line 33-34: Add reference for this sentence
  • Line 50-52: Add reference
  • Line 56: Replace “direct” with “directly”
  • Line 58: I suggest you remove reference #14, as it is not showing the impact of dense urban forms on creating local cool islands. This reference mainly explains the development of a web-based tool that calculates the Sky View Factor (SVF)
  • Line 64-66: Rewrite the sentence as “however, urban morphology has a significant influence on the thermal environment when the urban 63 compactness is higher than 0.15.”
  • Line 69: You cannot use the abbreviation H/W without explaining what it means. Replace H/W with height-to-width
  • Line 70: Why the author’s name (reference 18) is in all capital? Change it and also change the reference on page 18 accordingly.
  • Line 71: Replace “WRD” with “Weather Research and Forecasting Model (WRF)”
  • Line 72-73: What do you mean by “city’s urban center”?
  • Line 70-75: The entire sentence is vogue. You have to explain that they used two different scenarios (dispersed-city and compact-city scenarios) and then what they have found.
  • Line 85-90: One sentence; rewrite it into 2-3 sentences and explain the results clearly
  • Line 107: investigated
  • Line 109-112: Add reference

Study area

  • Line 132-133: I suggest you delete the sentence as you have the same sentence in the introduction
  • Figure 1: what ZC and ZD stand for?
  • Line 136-141: Add references for city info including the area, population, climate
  • Figure 2: Add the reference for the data used in this plot and explain the average annual temperature for which period?
  • Line 154-163: Were there specific reasoning for the Er-qi commercial district? Or just selected randomly?

Methodology and settings

  • Line 193-194: Add explanation on why those time frames were selected
  • Line 209: why that specific point was selected as the fixed point?

Methodology and settings

  • What method did you use to validate your collected data?

Results

  • Did you find a reason why humidity was higher in the fixed reference point? Probably add a little about the location of the reference point.
  • Add one sentence explaining the definition of “Air Temperature Ratio” to the manuscript

Discussion

  • As the main objective was to see if cities in cold regions act differently (vs hot regions), It would be great if you can add studies that did the same analysis in hot regions and discuss how the parameters influencing would be different

Conclusion

  • Change the title to “Summary and conclusion” as it is mainly a summary of the study

Figures

Figure 8: Suggestion; show the morning and night GCR plots in two different plots and prepare a new image with 4 plots in 2 rows and 2 columns

Figure 9: Same as figure 8, plot c is very busy

Figure 11: plots are very busy, make them more readable

Author Response

Reviewer2:

his study tries to analyze the influence of urban morphological indexes on temperature and RH changes in the morning, afternoon, and night in residential and commercial properties in Zhengzhou, China. This city is located in cold regions and the interest is to find out if the urban morphological indexes act differently in cold vs hot regions.

Review questions:

  • In-text citations have different formats. Some include the first name, some are all in the capital, etc. Check and rewrite all the in-text citations.

Response to the comment: Thank you for your kind suggestion. We have revised to family name only way.

  • There are some writing issues; you need to edit the draft. Make long sentences smaller so you can better explain matters.

Response to the comment: Thank you for your kind suggestion. We have revised some long sentences shorter.

  • Add the definition of Plot Ratio (PR) to the manuscript

Response to the comment: Thank you for your kind suggestion. Plot Ratio (PR) is located in last paragraph of introduction.

Introduction

  • Line 31-33: The sentence is not fluent.

Response to the comment: Thank you for your kind suggestion. This sentence has been revised.

  • Line 33-34: Add reference for this sentence

Response to the comment: Thank you for your kind suggestion. Reference for this sentence has been added.

  • Line 50-52: Add reference

Response to the comment: Thank you for your kind suggestion. Reference for this sentence has been added.

  • Line 56: Replace “direct” with “directly”

Response to the comment: Thank you for your kind suggestion. It has been revised.

  • Line 58: I suggest you remove reference #14, as it is not showing the impact of dense urban forms on creating local cool islands. This reference mainly explains the development of a web-based tool that calculates the Sky View Factor (SVF)

Response to the comment: Thank you for your kind suggestion. Reference #14 has been removed.

  • Line 64-66: Rewrite the sentence as “however, urban morphology has a significant influence on the thermal environment when the urban 63 compactness is higher than 0.15.”

Response to the comment: Thank you for your kind suggestion. This sentence has been rewritten.

  • Line 69: You cannot use the abbreviation H/W without explaining what it means. Replace H/W with height-to-width.

Response to the comment: Thank you for your kind suggestion. It has been revised.

  • Line 70: Why the author’s name (reference 18) is in all capital? Change it and also change the reference on page 18 accordingly.

Response to the comment: Thank you for your kind suggestion. It has been revised.

  • Line 71: Replace “WRD” with “Weather Research and Forecasting Model (WRF)”

Response to the comment: Thank you for your kind suggestion. It has been revised.

  • Line 72-73: What do you mean by “city’s urban center”?

Response to the comment: Sorry for the mistake we made. It has been revised.

  • Line 70-75: The entire sentence is vogue. You have to explain that they used two different scenarios (dispersed-city and compact-city scenarios) and then what they have found.

Response to the comment: Thank you for your kind suggestion. It has been revised.

  • Line 85-90: One sentence; rewrite it into 2-3 sentences and explain the results clearly

Response to the comment: Thank you for your kind suggestion. It has been revised.

  • Line 107: investigated

Response to the comment: Thank you for your kind suggestion. It has been revised.

  • Line 109-112: Add reference

Response to the comment: Thank you for your kind suggestion. Reference for this sentence has been added.

 Study area

  • Line 132-133: I suggest you delete the sentence as you have the same sentence in the introduction

Response to the comment: Thank you for your kind suggestion. The sentence has been deleted.

  • Figure 1: what ZC and ZD stand for?

Response to the comment: Sorry for the mistake we made. It has been revised.

  • Line 136-141: Add references for city info including the area, population, climate

Response to the comment: Thank you for your kind suggestion. References has been added.

  • Figure 2: Add the reference for the data used in this plot and explain the average annual temperature for which period?

Response to the comment: Thank you for your kind suggestion. References has been added. Dataset source of Figure 2 is from http://www.weather.com.cn/forecast/history.shtml?areaid=101180101&month, supplied by China Meteorological Administration. And we have added this information in Figure2.

  • Line 154-163: Were there specific reasoning for the Er-qi commercial district? Or just selected randomly?

Response to the comment: We selected Er-qi commercial district, because it is one of the biggest and oldest commercial district in Zhengzhou. It has a regular shape and the land use function is clear and definite. So we selected this district to do the measurement.

 Methodology and settings

  • Line 193-194: Add explanation on why those time frames were selected

Response to the comment: Thank you for your kind suggestion. Explanation has added in the first paragraph of section 2.2.

  • Line 209: why that specific point was selected as the fixed point?

Response to the comment: Thank you for your kind suggestion. The specific point is as an reference point to unify the diurnal changes among all the sampling districts.

 Methodology and settings

  • What method did you use to validate your collected data?

Response to the comment: Actually, we took two temperature and humidity auto meters from two different manufacturers, to make sure the measurement results are correct. Besides this method, we also validate measurement data with meteorological data supply by Zhengzhou meteorological bureau (http://zzqx.zhengzhou.gov.cn/).

Results

  • Did you find a reason why humidity was higher in the fixed reference point? Probably add a little about the location of the reference point.

Response to the comment: The reference point is located in a Foreign Language Primary School, the greenery coverage ratio is higher than the other area, this could be one reason that its humidity was higher than the other points. And the location of the reference point is added.

  • Add one sentence explaining the definition of “Air Temperature Ratio” to the manuscript

Response to the comment: Thank you for your kind suggestion. The explanation is added in the article (Air temperature ratio and humidity ratio refer to air temperatures and humidity for the points of the measurement routs divided by air temperatures and humidity of reference points).

Discussion

  • As the main objective was to see if cities in cold regions act differently (vs hot regions), It would be great if you can add studies that did the same analysis in hot regions and discuss how the parameters influencing would be different

Response to the comment: Thank you for your kind suggestion. References have been added.

Conclusion

  • Change the title to “Summary and conclusion” as it is mainly a summary of the study

Response to the comment: Thank you for your kind suggestion. Revision has been made.

 Figures

Figure 8: Suggestion; show the morning and night GCR plots in two different plots and prepare a new image with 4 plots in 2 rows and 2 columns

Figure 9: Same as figure 8, plot c is very busy

Figure 11: plots are very busy, make them more readable

Response to the comment: These are quite good suggestions. We tried to divide to smaller plots, however, the number of the plots are quite different. Fig.8 has 4 plots, while Fig. 9 and 11 have 5 and 8, respectively. Therefore, we didn’t revise these 3 figures this time. If insist on modify these figures we will try to find a better way for clearer logic and more regular form.

Round 2

Reviewer 2 Report

I still want them to answer the following questions:

Methodology and settings

Line 209: why that specific point was selected as the fixed point?

Methodology and settings
• What method did you use to validate your collected data?

Discussion
• As the main objective was to see if cities in cold regions act differently (vs hot regions), It would be great if you can add studies that did the same analysis in hot regions and discuss how the parameters influencing would be different

Author Response

Methodology and settings

Line 209: why that specific point was selected as the fixed point?

Response to the comment: That point is in a school campus near the measurement routes. It is selected because the point is inside the school and far from city main roads. In another words, it has a relatively stable environment. Moreover, the point is near the city center and near our routes, so it is selected and used to nondimensionalize data.

Methodology and settings
• What method did you use to validate your collected data?

Response to the comment: Firstly, we use 2 different measuring instruments to avoid instrumental error. Secondly, we also collected fixed point data and meteorological station data, and used them to do the validation.

Discussion
• As the main objective was to see if cities in cold regions act differently (vs hot regions), It would be great if you can add studies that did the same analysis in hot regions and discuss how the parameters influencing would be different.

Response to the comment: Thank you for your suggestion. We have added some references from line 658 to line 683.